# Double Discourse: Qualitative Perspectives on Breast Screening Participation among Obese Women and Their Health Care Providers

**DOI:** 10.3390/ijerph16040534

**Published:** 2019-02-13

**Authors:** Kate A. McBride, Catharine A.K. Fleming, Emma S. George, Genevieve Z. Steiner, Freya MacMillan

**Affiliations:** 1Translational Health Research Institute (THRI), Western Sydney University, Penrith, NSW 2750, Australia; catharine.fleming@westernsydney.edu.au (C.A.K.F.); e.george@westernsydney.edu.au (E.S.G.); g.steiner@westernsydney.edu.au (G.Z.S.); f.macmillan@westernsydney.edu.au (F.M.); 2School of Medicine, Western Sydney University, Penrith, NSW 2750, Australia; 3School of Science and Health, Western Sydney University, Penrith, NSW 2750, Australia; 4NICM Health Research Institute, Western Sydney University, Penrith, NSW 2750, Australia

**Keywords:** obesity, breast screening, mammograms, health service utilization

## Abstract

Obesity in Australia is rising rapidly, and is a major public health concern. Obesity increases the risk of breast cancer and worsens associated outcomes, yet breast screening participation rates in Australia are suboptimal and can be lower in higher risk, obese women. This study qualitatively explored barriers to breast screening participation in obese women in Australia. In-depth interviews (*n* = 29), were conducted with obese women (body mass index ≥ 30) and key health providers. A disconnect between providers’ and women’s perceptions was found. For women, low knowledge around a heightened need to screen existed, they also reported limited desire to prioritize personal health needs, reluctance to screen due to poor body image and prior negative mammographic experiences due to issues with weight. Providers perceived few issues in screening obese women beyond equipment limitations, and health and safety issues. Overall, weight was a taboo topic among our interviewees, indicating that a lack of discourse around this issue may be putting obese women at increased risk of breast cancer morbidity and mortality. Consideration of breast screening policy in obese women is warranted. Targeted health promotion on increased breast cancer risk in obese women is required as is a need to address body image issues and encourage screening participation.

## 1. Introduction

Breast cancer is the most common cancer among women globally, with over two million new cases in 2018 [1]. In Australia, which has the seventh highest rate of breast cancer worldwide, eight women die each day in due to breast cancer, and it is the second leading cause of cancer-related deaths among women in the country [2]. With one-quarter of hospitalizations occurring for Australian women due to breast cancer, the disease places a heavy burden on the healthcare system. Expenditure on breast cancer treatment, which currently totals approximately $411 million AUD annually, is also forecast to rise due to increasing incidence in coming years [3,4]. This increase is, in part, attributable to a growing aging population and increased detection by screening. While overdiagnosis and over screening among average risk women are leading to both growing health concerns and areas of research [5,6], it is still important to identify those most at risk to optimize their screening participation through targeted interventions. 

Environmental factors can influence breast cancer risk and outcomes, with high body mass (BMI) being a leading modifiable risk factor for post-menopausal estrogen receptor breast cancer [7,8,9,10]. Obesity is associated with more aggressive clinical presentations and substantially worse outcomes for all breast cancer subtypes [8,11,12,13,14]. This includes being an adverse prognostic factor in response to adjuvant chemotherapy [15], and contributing to higher mortality rates in association with breast cancer [16,17]. Obesity rates are increasing globally with an estimated 640 million adults obese in 2014 and an age-standardized prevalence of 14.9% of women [18]. Australia experiences some of the highest overweight and obesity rates in the world. In 2014–2015, approximately 64.3% of the Australian population were either overweight or obese., with 27% of the population were obese in the same time period [19]. In certain urban areas, such as Western Sydney which also has high rates of overweight/obesity (51.7%) [20], rates of breast screening are as low as 47.5%, well under the nationwide target of 55% [21], suggesting that a proportion of eligible higher-risk women are not participating in recommended breast screening. This growing trend is concerning given that obese individuals have been shown to be less likely to participate in preventative health behaviors like mammographic screening [22].

Low levels of mammographic screening participation have been associated with obesity among women in several settings, largely established through self-reported screening behaviors [23,24,25,26,27]. Only one Australian study to date found that obese women are 8% less likely to screen than normal weight women, but again this was self-reported screening data, likely exaggerating the proportion of women actually screening [28]. Body image disturbances, such as body shame and body avoidance, appear to play a role in frequency of cancer screening participation among women. A number of quantitative studies have found that these disturbances may affect participation in screening, though this relationship has not been specifically explored in obese women [29,30,31,32]. 

Some qualitative data also exists around barriers to cancer screening among women, though much of this research focusses on body image in women of varied weight. For example disgust with one’s own body has been found to be a key motivation in avoiding cervical screening, with the fear of having a stranger see their body overriding these women’s perception of cervical cancer risk [33]. To our knowledge, only one study to date, conducted in the United States of America (USA), has explored barriers to mammographic specifically among obese women [34]. That study identified a number of barriers common to the general population such as fear, modesty, and low perceived cervical and breast cancer risk. Weight related barriers were also found, and included experiences with equipment that could not accommodate the women and insensitive comments being made by technicians and health professionals about weight. To date no such study has been conducted in Australia, arguably a different setting due to differences in both the healthcare system and availability of a national Breast Screen service, which provides fee-free screening for women over 50 years by female radiographers. Healthcare professional provider perspectives (including those of mammographic staff like radiographers) have yet to be explored.

There is a need for research to explore reasons from both the perspectives of women and key providers as to why women at higher risk may or may not attend breast cancer screening, in order to develop targeted and appropriate interventions. The aim of this study is to identify facilitators and barriers to breast screening participation in obese women and investigate key health care provider perspectives on service provision for this population in an Australian context. 

## 2. Materials and Methods

Participants were women from the Western Sydney region of New South Wales (NSW), Australia who self-identified as being obese (BMI ≥ 30 kg/m^2^). Women from age 45–80 years were targeted. While women are only invited for biennial mammograms in Australia from 50–74 years, women are able to self-nominate for a free mammogram from 40 years, as are women older than 74 therefore may have had recent mammogram. Recruitment occurred via social media, local community events, and through key community facilities, such as shopping centers. Consented participants were then invited to participate in semi-structured interviews via telephone. 

A semi-structured interview schedule (Appendix A) was developed by a multidisciplinary project team (public health, psychology, health promotion). The interview schedule included a series of key open-ended questions developed from a previous study that examined perspectives of cancer screening among high hereditary risk individuals [35]. The schedule began with a general question on breast screening participation and this then determined the questions that followed for screeners and non-screeners. Non-screeners were asked about prior screening experiences, possible barriers to their attendance at screening, and possible facilitators to increase their screening. Existing screeners were asked about their experiences of screening, the factors they think contribute to a good or bad experience of screening, and the barriers and facilitators to screening participation.

One-on-one interviews with de-identified key healthcare providers such as mammographic staff and service providers across healthcare organizations were also conducted. Providers were invited to participate in semi-structured face-to-face or phone interviews with a trained researcher (CAKF) and were asked about their perceptions on service delivery and how this might impact on breast screening participation. Mammography staff were also asked what the experience of conducting a mammogram was like for obese clients, the factors and circumstances that may affect women’s perceptions of the examination, and their experiences and perceptions of clients who are obese (Appendix A).

Interview responses were digitally recorded and transcribed verbatim by a professional transcription service. An inductive coding approach as previously outlined [36], which analyzed provider and female participant data separately, was utilized for data analysis through the qualitative software Quirkos [37]. This software allows for transcripts to be read and the text to be considered for the multiple meanings which were inherent in the body of the interview text [36]. Text segments containing meaningful themes relating to study objectives were assigned to categories by a member of the research team (CAKF). Throughout the analysis process, a hierarchy of categories was then established to show various relationships between categories [36]. Continued revision of the categories and emerging themes then took place with the reviewer searching for sub-topics, contracting points and new insights into each category. To demonstrate the emerging themes and categories identified during analysis, tables under each category have been included in the results section with illustrative quotes for each sub-topic. Demographic data (e.g., age, screening status etc.) was collected to describe the characteristics of the sample. Fifteen per cent of interview data were coded independently by two research team members investigators (C.A.K.F., K.M.B. (Wolf, Managing incidental findings and research results in genomic research involving biobanks and archived data sets)), with an inter-rater reliability of 87% agreement reached. Ethical approval for this study was provided by Western Sydney Human Research Ethics Committee (H11725).

## 3. Results

A total of 19 obese women, mean age 52 (Table 1), and 10 providers (Table 2) agreed to participate in the study, with interviews conducted in 2017 and early 2018. Each interview lasted on average 45–60 min. Recruitment was ceased once theoretical saturation was reached. 

Emergent themes from the obese female participant data were sorted into three main categories, all containing several sub-categories: (1) Obese women’s understanding and awareness of screening can affect participation; (2) Body image concerns among obese women impact on screening attendance; (3) Negative experiences for obese women during screening can act as a barrier to future screening. Themes emerging from the provider data were sorted into two categories, again containing several sub categories: (1) Provider reported experiences with obese screeners; (2) Providers do not see obesity as being a barrier to breast screening. Examples of obese female participant and provider excerpts for each of these emergent themes can be found in Table 3, Table 4, Table 5, Table 6 and Table 7. 

### 3.1. Obese Women’s Understanding and Awareness of Screening can affect Participation

There were a number of factors identified by our participants as reasons to take part or not take part in breast screening. A family history or personal experience of breast cancer were identified by several participants as reasons to participate in breast screening (excerpts 1.1 and 1.2). For others, however, even a family history and encouragement by other family members was not enough for them to prioritize screening (excerpt 1.3). One participant also stated a lack of family history had led to her conscious decision to not participate in routine breast screening.

Further, multiple participants said they did not understand why screening was a necessary or a positive health behavior. Only one participant was aware of her increased risk of post-menopausal breast cancer making reference to her weight (and her personal history of smoking and drinking), (excerpt 1.4). Several women lacked health prevention literacy around mammographic benefits and harms—‘mammograms give you cancer,’ with these misconceptions proving to be a further barrier to screening. Other women still screened, however, despite similarly not being educated or aware of the purpose or effectiveness of breast screening, or even if it still viewed as a secret issue (excerpts 1.5 and 1.6). Participants described positive influences on their screening behavior, such as the media reporting breast cancer-related deaths of well-known Australian public figures (excerpt 1.7) and encouragement by their general practitioner to screen, even in women who were busy with other life events (excerpt 1.8). Important persons in the lives of the participants were also seen as encouragers of screening by either providing group screening support (excerpt 1.9), or by giving women a reason to look after their health such as one woman described, so she could ‘be around’ for her young daughter (excerpt 1.10).

Prioritization of screening was an issue for several of the women we spoke to, with lack of time, laziness, and simply having greater priorities (excerpts 1.11 and 1.12). Fear of pain was also a commonly cited issue in regard to follow up breast screening after an initial screening event, with this pain thought to be exacerbated by having larger breasts (excerpt 1.13).

### 3.2. Body Image Concerns among Obese Women Impact on Screening Attendance

Feelings of self-consciousness and body image concerns contributed to their negative experiences and perceptions of breast screening, which subsequently led to increased reluctance or avoidance of the procedure. Body image concerns due to a higher BMI were commonly cited as being a contributor to a reluctance to screen among both never and lapsed screeners in this study. Breast screening was avoided by some as they felt it reminded them of never feeling good about their bodies (excerpts 2.1 & 2.2). 

This self-consciousness about being confronted by one’s own body was exacerbated by what was already seen as being an unpleasant experience (excerpt 2.3), with having to see one’s large breasts being ‘squashed’ considered to be ‘the last thing’ these women would choose to do (excerpts 2.4 and 2.5).The sensitivity of the radiographer appeared important in how body image concerns were managed and could both negatively and positively impact on the screening experience. For women who ‘did not feel okay about their body,’ lack of sensitivity and communication about how and why their breasts were handled during the procedure added to their poor experience and further obstructed future screening attendance (excerpt 2.6). Positive experiences were also reported, however, with these largely attributed to the manner and communication of the radiographer (excerpts 2.7 & 2.8). 

### 3.3. Negative Experiences for Obese Women during Screening can act as a Barrier to Future Screening

Screeners in this study reported both negative and positive breast screening experiences in association with their weight. Negative experiences were both physical (largely due to the size of the breast) and psychological. Physically, mammograms could be extremely painful due to large breast size, and uncomfortable due to an inability to get close to the plate because of body size in general (excerpt 3.1). Adverse psychological events were also because of this ‘manhandling,’ with one woman reporting this could trigger off her past experiences of sexual assault (excerpt 3.2). Another negative psychological experience reported among women who had screened was around having anxious thoughts, thinking something had been found during their mammogram. This was attributed to minimal communication by the radiographer on the additional images required, due to their larger breast size. Negative body image perceptions also impacted on mammographic staff, with weight contributing to a negative experience for the radiographers (excerpt 3.4 & 3.5).

### 3.4. Provider Reported Experiences with Obese Screeners

Impacts on the procedure as well as screening length due to the size of the women being screened. These were common reoccurring themes with the majority of providers during their interviews, which sometimes led to a focus on the task, rather than the patient. For example, limitations with available equipment caused issues during mammogram with the ability of the machine to reach the desired compression reported to be compromised for patients with larger breasts (excerpt 4.1). Positioning of the patient was reported as being problematic, often due to the patient having a large stomach and the need for an anatomically inferior view (excerpt 4.2). Size also led to an increased number of images being required (excerpt 4.3) which in turn increased the ‘manhandling’ required, putting patients at increased risk of adverse events such as splitting the skin under the breast (excerpt 4.4). A typical outcome of these issues would be for patients having to subsequently return to complete their mammogram, sometimes at another clinic. The number of images required was not the only factor that could increase the mammogram length. Obese patients reportedly can become short of breath, limiting the amount of time they can stand, meaning that breaks also need to be taken to give the patient a rest (excerpt 4.5).

Issues around obese women were also reported for mobile screening vans, where one obese patient reportedly had to be hoisted on the wheelchair lift into the van as she could not walk up the narrow stairs. Providers also felt that the limited size of the van waiting room, change area, and walkways could be problematic for obese patients (excerpt 4.6). Equipment issues were not limited to impacts on the screening procedure, or patients’ comfort. Concerns around work health and safety were also reported, with time and consideration needed when positioning large breasts to avoid injuries (excerpt 4.7). Regardless of these experiences, the providers we interviewed felt that discussing a patient’s weight with them was unacceptable.

This meant that sometimes, the identified unforeseen technical and safety considerations became the priority and could consume a considerable amount of the radiographer’s focus to ensure that satisfactory images were obtained, rather than consideration of the women themselves (excerpt 4.8). Other radiographers described trying to be ‘lenient with first timers’ with regards to compression while still maintaining image quality (excerpt 4.9), with speed the priority over patient comfort.

### 3.5. Providers do not See Obesity as Being a Barrier to Breast Screening

The majority of providers felt that size and/or weight was not associated with additional negative feelings or fears towards accessing breast screening services unless it was a co-morbidity alongside a mental health condition (e.g., body dysmorphic disorder) (excerpts 5.1, 5.2 & 5.3). Instead providers felt other barriers, such as cultural barriers and a lack of education were the main drivers of low participation among women like those interviewed as part of this study (excerpts 5.7 & 5.8).

Providers also felt that weight was a taboo topic with a strong reluctance to identify weight prior to booking, despite the equipment limitations discussed above. Providers underestimated how their female patients felt about their weight—mistakenly assuming women simply had a problem with screening due to a fear of being judged because of their weight (excerpt 5.9). 

## 4. Discussion

Our study is the first to qualitatively explore breast screening participation among both obese women and key breast screening providers. Undertaken among obese women living in Western Sydney (WS), Australia as well as key breast screening providers from the same area, our study identified a number of barriers and facilitators of mammographic breast screening. Many of the barriers identified were common to women in the general population, such as perceptions of being at low risk (despite being at higher risk due to their weight), low understanding around benefits and harms of screening as well as low priority of breast screening [38,39]. Common encouragers were positive social influences like family and high public profile figures, encouragement by primary care providers, awareness of heightened risk due to family history of breast cancer, and a positive previous experience at screening [1]. In addition to the identified common barriers and facilitators, issues unique to this population of women with a higher BMI and their mammographic breast screening behavior were found.

Feelings of self-consciousness and embarrassment about revealing their bodies were identified by obese women as contributing to their negative perceptions of breast screening, which subsequently, led to reluctance or avoidance of screening. Not feeling positive about their body led to inferior screening experiences by the women, which was also compounded by a lack of sensitivity on behalf of screening staff. This experience is consistent with findings from previous research about mammographic breast screening [34], as well as research examining obese women and their cervical screening behaviors [40]. That empirical evidence found that insensitive or offensive comments made by staff were additional barriers to screening in those populations Further understanding of whether these body image concerns are primarily due to larger size or whether it impacts on mammographic screening by women of all body size is needed. Increased pain, perceived by our participants as being due to a larger breast size, was also a strong discourager of future screening, again consistent with previous research [34,40]. 

A further novel finding of this study, was that a larger breast size also increased physical manhandling during mammograms, which made participants feel more uncomfortable and body conscious during their screen. Lack of communication from the mammographic staff appeared to enhance participants’ feeling of discomfort, indicating that staff involved in the mammographic process may need to consider how to communicate the need for repeated repositioning of the breast. Women perceived that they had increased anxiety levels due to the limited communication around the length of screen, as staff failed to inform the women they were having multiple images taken because of their large breasts (and not because something had been detected on imaging). This finding is pertinent given that the mammographic staff who took part in this study reported that their focus was on obtaining the right image rather than ensuring the women had an acceptable experience of having the mammogram. This poor practitioner-patient communication is consistent with previous research among obese women and their cervical screening behaviors [40], and may be due to weight being seen as a taboo topic. Research examining the communication needs of obese women during their mammogram appears warranted.

The current study also explored the perspectives of key providers involved in mammographic breast screening. Several novel findings came to light, including the additional burden on providers due to equipment limitations, difficulties in positioning obese patients, work health and safety issues, extended screening times, and higher risk of adverse events. While none of the providers interviewed insinuated any negativity towards screening obese women, the identified increased work load for radiographers, in particular, has the potential to lead to strained patient/practitioner relationships. This may be exacerbated by weight not being identified or discussed prior to booking or during the mammogram as standard practice. The providers in this study were reluctant to be assertive or approach weight as being a health condition, leading to added pressure and stress on radiographers when performing procedures. It appears that some health practitioners may be leaving patients to identify themselves as being obese, information many women may not be forthcoming with, or indeed be aware they are classified as being obese. An underlying fear also appears to exist among providers in this study that if women are identified or forced to identify as being obese during the booking process, they are less likely to attend the appointment. A straight forward strategy may also be to simply ask for height and weight as part of the booking process with BMI then automatically calculated to sidestep this issue. Overall, these findings suggest a need to develop and evaluate effective communication strategies for radiographers/booking officers to ensure improved identification of patients requiring: (a) more time, (b) specialized/larger machinery, and (c) more images taken. Further, development of practical strategies on how to conduct screening with obese women, including sensitivity training could reduce the negative experiences of obese women as well as improve screening length times.

Surprisingly, despite the identification of a number of issues for clinical staff, the providers we interviewed did not see obesity as being a barrier to breast screening. Instead, providers held either the view that women felt uncomfortable about screening in general, regardless of size, or that the reluctance of obese women to screen could be attributed to cultural and/or health literacy barriers. This is indicative of a lack of awareness of the barriers obese women may have to mammographic screening. Highlighting that, on comparison of both provider and obese women recounts, a double discourse exists between perspectives on how weight impacts on breast screening participation. For the majority of our providers, obesity was only an issue from their own practical work flow perspectives. Contrary to what we were told by our sample of obese women, providers did not see how feelings of discomfort during a mammogram may be amplified for obese women or that negative feelings around being obese could prevent women from attending a mammogram at all. Several of our providers suggested that overweight/obese patients did not face any additional psychological barriers (from their experience) despite our sample of obese women stating that they did, it may not have been externalized by the obese women they have screened previously. This understanding of barriers contradicts some of the existing literature, where clinicians thought that embarrassment, being unwilling to undress, and lack of desire to discuss weight may be reasons for obese women not wanting to screen [41]. This contradiction may exist as the clinicians in that study were family physicians who may encounter obese individuals more frequently, unlike the providers in this study. The findings may also be due to the: (a) providers in this study not being obese themselves, albeit observational data and not collected as part of the demographic data for this group (and therefore failing to recognize the stigma around body image obese women can experience), (b) because weight appears to be a taboo topic, and (c) because obesity appears to be an unrecognized barrier to screening in this setting, despite it being associated with non-attendance in the literature [28,29,42]. Given the identified barriers that obese women are facing for breast cancer screening, there appears to be a need to investigate further why weight is such a taboo topic for health professionals, and a need for education around communication with obese women.

As with all qualitative studies, this research has limitations and is intended to provide suggestions for future research directions, rather than establish causal relationships. There were a number of limitations specific to this study including self-reported weight and screening, an inability to compare obese and non-obese women as we did not interview non-obese women (as this was beyond the scope of this project), and being based in a setting where mammographic screening is freely available to women aged 50–69, therefore findings may not be transferrable to other settings.

## 5. Conclusions

While not all providers felt that size and weight were associated with any additional negative feelings or fears towards accessing services, obese women identified that negative perceptions of their body affected their attitudes and behaviors towards breast screening. There appears to be a lack of discourse around these issues regarding weight in breast screening services in Australia which can be generalized to other settings which offer mammographic screening, and may be affecting attendance at breast screening by obese women. In addition to targeted novel health promotion strategies to raise awareness of the increased risk of postmenopausal breast cancer due to a higher BMI, changes to clinical practice should be considered including open identification of weight prior mammogram appointments, referral of obese women to services with equipment appropriate for larger women, larger women advised of likely longer screening time prior to their mammogram and education for mammographic staff. Further research should also be conducted to create a comprehensive picture of breast screening patterns among obese women, using routinely collected data that can be leveraged to inform appropriate community and screening service-based interventions. Impacts on attendance to healthcare services in Australia in general may also be needed given obese women’s increased risk of multiple chronic diseases, not just breast cancer.

## Figures and Tables

**Table 1 ijerph-16-00534-t001:** Demographic characteristics of obese women and providers.

Obese Women Characteristics	*n* = 19
Mean age (range)	57 (47–75)
BMI ≥ 30	19 (100%)
Screening history yes/no	10/9

**Table 2 ijerph-16-00534-t002:** Provider characteristics.

Provider Characteristics	*n* = 10
Female Median length of service years (range)	10 (100%) 10 (4 to 21)
Provider categories	
Radiographers/clinical staff	5
Health promotion/bookings staff from a government-funded breast screening organization	3
Primary Health Network staff	1
General practitioner	1

**Table 3 ijerph-16-00534-t003:** Obese women’s understanding and awareness of screening can affect participation.

Emerging Theme	Excerpt Number	Excerpt
Family history	1.1	‘a friend who did breast screening and found that she did have breast cancer and she told my friends about the importance of it’ (Participant 14) *
	1.2	‘The funny thing is because I’ve got breast cancer in my family, I check myself for breast cancer all the time, but I haven’t had a Pap’s smear in God knows how long, which just sounds stupid because it’s the same, cancer is cancer, but I don’t, I only—I don’t check myself for that’ (Participant 17)
	1.3	‘Yeah, I’ve talked about screening with my mother and with other relatives. I’ve got an auntie who had breast cancer and she picked it up by self—through self-examination, but she was 60, had never had a breast screen and so she’s like go and get your boobs checked. Yes, yes, I will, and I’ve just never—it’s just not top of the mind.’ (Participant 2)
Understanding benefits of screening	1.4	‘Yeah, from the—you know, from the public health sort of perspective, so getting in early. It’s important to me like sort of socially but you know, for health of Australians, but also personally obviously, partly because I’m you know, reasonably well-educated so I understand that my weight will—and I’ve had a history of smoking and drinking, so all of those things put together will increase my—and poor exercise history as well, so all of that together says that I absolutely should screen.’ (Participant 1)
	1.5	‘I wasn’t given any information. I was just like given a piece of paper and sent to a radiology place.’ (Participant 10)
	1.6	‘I think there’s still a certain amount of secrecy with it with women, which is why I’ve always discussed it with my daughter and my granddaughter. It’s just another part of your body that’s got to be checked.’ (Participant 3)
Screening influencers	1.7	‘It was highlighted by the deaths of two people that were in—not in my life but Jane McGrath and Rove’s wife. They were the two.’ (Participant 8)
	1.8	‘So having a GP that was there to say righto, it’s time for you to maybe have a look at this and have a look at that, that was a kind of a good thing for me because, as I said, my life has been choc-a-block of kids and everything like that and work, that I probably would have overlooked it and you know, I’d not have got that leeway, at least until we lost my mum or my mum got sick, and then we were kind of all into everything about it.’ (Participant 8)
	1.9	‘I mean there’s a lot of concurrence. A lot of friends don’t enjoy it either. Most of my friends go together. I know a lot of my friends have the thing of going together to get screened so that they do—(a) so that they do it, but also that they can provide each other with support.’ (Participant 1)
	1.10	‘It’s my daughter though, because a single mum and she doesn’t have anyone else, so I need to look after my health because I need to be around for a lot longer. She’s only 12.’ (Participant 2)
Lack of priority	1.11	I feel—the first thing that jumped into my head when you asked me that question is I need to clean my oven. Well you know, and if I don’t well it’s just inevitable it’ll have to happen at some point, I’ve got to do it, and so it’s like that. But it’s a—it’s not something I like doing at all but I feel like I’ll get into doing it’ (Participant 1)
	1.12	‘No, not really. No, I’m just lazy. I’m really time poor and so getting around to making the appointment and having you know the screen done. It’s like I’ve got the bowel cancer screen sitting at home waiting to be done. That was mailed to me when I turned 50. I know how important they are, I know people who received it in the mail and then been picked up and it saved their life. I know how important screening is. I’m very time poor and I’m just—I need to get myself organized to get it done.’ (Participant 2)
Fear of pain	1.13	‘The mammogram was extremely painful. I’m a large lady and also have large breasts, and honestly, I felt—not to be gross, but it was like them trying to pop a zit is how it was for my breast…to be honest, it’s put me off having another one, unless I actually find something wrong’ (Participant 19)

* Participants are from the obese women’s group.

**Table 4 ijerph-16-00534-t004:** Body image concerns impact on the screening experience for obese women.

Emerging Theme	Excerpt Number	Excerpt
Avoidance of screening due to body image concerns	2.1	‘I don’t ever feel good about my body and I’m reminded about that when I have a breast screen’ (Participant 6)
	2.2	‘So it is a big thing for me to try and get under control and—but it also is something that I kind of—probably if you asked me how’s your life going, I’m sad I’ve lost my mum but that’s only been for three years, but my saddest part of my whole life has been my weight, so it’s been something that’s bothered me all the time.’ (Participant 8)
Being self-conscious exacerbates unpleasant screening	2.3	‘I was saying before, just to experience the whole thing visually, to start hating on yourself about that whilst in pain and there’s somebody that’s not—this is a bad experience’ (Participant 4)
	2.4	‘There’s also some self-consciousness when you’re overweight, actually—to be honest, actually watching your breasts squashed under a plate is—it’s about the last thing I’d choose to do (Participant 1)
	2.5	It didn’t stop me going. Certainly, no one want to take of their clothes and get squeezed in between some cold metal plates and stand in an awkward position but it didn’t stop me from going. It wasn’t the nicest experience, but it was something that you just have to do’ (Participant 17)
Radiographer sensitivity	2.6	Yeah, a person handling, manhandling your breasts into place, … not every practitioner that I’ve met, not every mammographic that I’ve met had that, had that sensitivity…one of them…I actually thought about making a complaint, and then I thought look, there’s not really any complaint that I can make. He needs to get my breast onto the plate. You know, the fact that he’s just acting like a bit of a pig is off but it’s not enough to actually—you know, to put a complaint in anywhere. But certainly, you know, the sensitivity for what it might be like for women who don’t feel—like I say, women who don’t feel okay about their body (Participant 1)
	2.7	‘No, no, no, they’re all very pleasant. I mean it’s not a nice thing to have done, but all the women I’ve ever dealt with have been very pleasant, very amiable, never made you feel embarrassed.’ (Participant 3)
	2.8	‘my more recent encounters are more positive, that it’s really great you’re here… That was all very positive and thanks for coming and participating. It’s just been a much more in positive encounter’ (Participant 18)

**Table 5 ijerph-16-00534-t005:** Negative experiences for obese women during screening act as a deterrent to future screening.

Emerging Theme	Excerpt Number	Excerpt
Negative physical and psychological events	3.1	‘I’ve spoken to a couple of friends of mine who also have had mammograms and especially the bigger-breasted women, they say they have the same problem. It’s very uncomfortable for them. And not just the bigger-breasted but the bigger in size too because they want you to get your chest so close to the plate, but if you’re overweight like I am, sometimes it’s very hard to actually get your body there.’ (Participant 19)
	3.2	‘If I can say something else, I hadn’t actually imagined that I’d be sort of hoisted left, right and center to get my breast onto the plate, and so again that can trigger off my you know, past experiences being the sexual assault and so it can actually trigger off unwanted contact in the past because it is actually somebody—I would never have a male mammographer again, not that I had an issue with it on the way in, he was such a jerk that it put me off for good.’ (Participant 1)
Perceived impact on the radiographer	3.3	‘No, no, not really. I think—no. I think that you—in my head I just accept the fact that you know, I’m one of the fat ones, that’s all, but then I give them a lot more to work with, so they’re really pushed to the edge. I make sure they can do it properly. Anyone can get a little boob in there. You try and get this big one in.’ (Participant 8)
Positive experiences	3.4	‘No, no, no, they’re all very pleasant. I mean it’s not a nice thing to have done, but all the women I’ve ever dealt with have been very pleasant, very amiable, never made you feel embarrassed.’ (Participant 3) ‘Because I’ve put on weight, so I’ve got more breasts for them to play with when I’m having a mammography which makes it harder for them… That’s what I’ve put it down to, they’re bigger.’ Participant 19

**Table 6 ijerph-16-00534-t006:** Provider reported experiences with obese screeners.

Emerging Theme	Excerpt Number	Excerpt
Patient size impacts on mammogram efficiency and safety	4.1	‘Yeah, so there’s one unit that has limitations… the compression has to come down to a certain point if we can take the image, so if the lady’s got quite large you know, voluptuous breasts, then that compression can’t come down enough to be able to enable the exposure. So yeah, I’ve had to on occasions just say look, I can’t do it, or if it’s sort of—if they’re borderline then I’ll say okay, next time maybe go to RPA because this is what’s happened today, and to avoid that next time just go to a different site.’ (Provider 3)
	4.2	‘We do have trouble, we have only trouble positioning them, bringing them forward to the detector is when they have very protruding stomach. That’s where the problem is so they aren’t able to really lean forward, especially when we go into the different view, and then we have to get the angle at the bottom here and then the tummy’s in there, so that’s when we find it really difficult to position them and get a good radiograph, yeah.’ (Provider 6)
	4.3	‘So, we split it up, so we just look at the top part of the breast and let the tummy be there, and then we turn the angle a bit and position a little bit different to get the lower part of the breast, yeah.’ (Provider 6)
	4.4	And as well as—they usually have very thin skin underneath and sometimes when—just when you pull it splits, the skin splits, and so that is one difficulty. So, it’s sometimes very difficult to carry on and we have to just stop the procedure and get them to come on another day because it gets painful, so that’s another thing we come across as well, yeah.’ (Provider 5)
	4.5	‘Technically it’s difficult. Sometimes when they’re obese they’re short of breath as well, so we have to get them to sit in between views, yeah, that too.’ (Provider 5)
Obese patients cannot use mobile screening vans	4.6	‘No, I think it would just be in terms of the van, yeah. Our old van obviously when I had that issue with that lady, you know the new vans are slightly bigger, but I do think with the van the waiting room is quite small, the change room is very small, and the walkway I think if….They’re very small…..Yeah, so the walkway to get into the mammo room, for an obese lady I think it would be very difficult.’ (Provider 7)
Work health and safety	4.7	‘You just have to break it up, take your time, be conscious of how you’re moving. Often—because the breast can be quite heavy, so you’ve just got to be conscious of your movements so that you don’t injure yourself. I’m very, very much aware of that.’ (Provider 2)
Quality images can take priority over patient considerations	4.8	So just like speaking with—reassuring the that you know, I meet a lot of women and it’s not something I actually pay attention to, it’s just —it’s all about getting a good picture, making sure they don’t have to come back for extra views that are technically unsatisfactory, all of that.’ (Provider 3)
	4.9	‘And you know, for the first timers you know, you’re a little bit more lenient and just guide them through the compression a little bit more. And you know, the whole point again, the training comes in, how to handle patients, how to make sure that you get adequate compression to get good quality images comes into play and then you know, doing the procedure quickly.’ (Provider 5)

**Table 7 ijerph-16-00534-t007:** Providers do not see obesity as a barrier to breast screening.

Emerging Theme	Excerpt Number	Excerpt
Weight is not barrier to screening	5.1	‘Well no, not really. Now I mean we get you know, concerns from all sorts of women, skinny women, larger women. Yeah, that’s not one particular you know, concern that sticks in my mind and as I said you know, I’ve never really had anyone say to me they don’t want to come in because they’re obese’ (Provider 7)
	5.2	‘I have never—yeah, I’ve never encountered for example an obese woman saying that’s the specific reason why she’s not coming back, for example.’ (Provider 4)
	5.3	‘Look, I honestly don’t think that someone who’s overweight would be more apprehensive, except if they’d previously had a bad experience somewhere having a mammogram due to their weight problem, and so—but that’s no different in some ways to a client who has no breast tissue at all, and it’s extremely difficult for them as well. So, I really wouldn’t say that there’s any one category that’s more apprehensive unless it’s due to a mental condition. I would say that obviously there are some clients who are developmentally delayed who through that condition are extremely overweight, so you have that combination.’ (Provider 3)
Other perceived barriers	5.7	‘Cultural barriers as well because we’ve got a lot of different cultures. Arabic women you know, are often very reluctant to screen.’ (Provider 3)
	5.8	‘Yeah, I think it’s just you know, just education, more education that’s needed a lot of the time.’ (Provider 1)
Misconceptions of the impact of weight	5.9	‘It’s always a taboo subject, weight, isn’t it…? Maybe they think they’re judged because they are fat because—also overweight because maybe they think society might think that that’s not acceptable.’ (Provider 8)

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
