# Peer review of "Double Discourse: Qualitative Perspectives on Breast Screening Participation among Obese Women and Their Health Care Providers"

_ijerph, 2019, doi:10.3390/ijerph16040534_

Round 1

Reviewer 1 Report

This is an interesting manuscript that discusses the results of qualitative investigations into patient and provider perspectives on breast cancer screening among obese women in Australia. The authors conducted key informant interviews with 19 obese women and 10 clinicians/providers. They observed a disconnect between patient and provider observed barriers. For example, patients identified

I have several comments that the authors may consider addressing in revising this manuscript.

Would the authors consider using the term “provider”, “clinician” or “healthcare stakeholders” rather than just “stakeholders” for the clinical interviewees? The women themselves could also be considered stakeholders; the healthcare providers are not the only stakeholders in this discussion.

The introduction section is very long.  Recommend reducing if possible. Indeed, the whole manuscript is very long.  The authors should look for opportunities to shorten.

Would the authors consider providing the patient and provider interview schedules/questions as supplementary material?

Methods: were the interviewers transcribed by the interviewer or another researcher? How were they analysed (using what software, methods, etc?)

Methods: Can the authors (circa lines 125) provide any data on agreement between two coders?

Methods: Can the authors describe any theoretical framework used to analyse the data? It is not clear how the themes were identified. Were the data from patients analysed separately to those from providers?

Table 1. This table is unclear. Can the authors reorganise to show the patients and providers separately? Currently, the “stakeholders” are described under a table that is entitled “obese women”. The difference between the two is not clear. Do the authors have information on average/median length of clinical experience among the clinical stakeholders?

This reviewer found the results section very confusing.  I would like more differentiation between presentation and discussion of patient and provider results. It is not clear which of the themes were identified from patients and which from providers.

Results: The first “theme” (multiple encouragers and barriers…) seems more like a finding than a theme?  It seems many themes are discussed in section 3.1 and table 2. It is not clear how the themes presented in this section are different from the themes presented in the other sections? Thus, it was not clear to me how the items discussed in Tables 2,3, and 4 are different, and why they were separated as such?

Line 150.  The “unhealthy behaviours” are not mentioned in the methods section. Are these investigator assessed or self-reported/identified?

Is there a way for authors in the tables to distinguish which items refer to barriers and which facilitators?

The authors identify several body image concerns discussed by obese participants in the interviews. Are these the “result” of their obesity, or do non-obese women have the same concerns? This may be worth discussing in the context of the literature in the discussion section.

Section 3.4 The first two paragraphs seem to discuss redundant themes.  Can the authors merge and shorten?

Line 283: suggest revising the word “distain” to “negativity,” or similar. The very negative language appears implicitly fatphobic?

The authors discuss the need to understand the communication needs of obese women. However, I think it’s also worth discussion that providers need to be trained to conduct screening with obese women, including sensitivity training.  

I would like to see the authors mention next steps from this work in the conclusions section. What changes to clinical practice can be recommended as a result of this work? What gaps still exist?

Author Response

Response to reviewer one:

We thank the reviewer for their time in looking at this paper and providing such useful feedback to help us improve the manuscript. Please see our responses to the comments below.

Point 1: Would the authors consider using the term “provider”, “clinician” or “healthcare stakeholders” rather than just “stakeholders” for the clinical interviewees? The women themselves could also be considered stakeholders; the healthcare providers are not the only stakeholders in this discussion.

Response: We agree that the women we interviewed could also be considered as being stakeholders therefore have updated the term originally used to ‘providers’ throughout the manuscript

Point 2: The introduction section is very long.  Recommend reducing if possible. Indeed, the whole manuscript is very long.  The authors should look for opportunities to shorten.

Response: We agree that the introduction is quite long and have reduced it where we were able an have also reviewed the entire manuscript and made reductions were possible

Point 3: Would the authors consider providing the patient and provider interview schedules/questions as supplementary material?

Response: This is an excellent suggestion. We now include the interview schedule in supplementary file x

Point 4: Methods: were the interviewers transcribed by the interviewer or another researcher? How were they analysed (using what software, methods, etc?)

Response: We have added in the following detail on p169 “by a professional transcription service”. However in terms of analysis and software etc, this detail is already in the manuscript (lines 170-182) and which reads as follows: “An inductive coding approach as previously outlined [39] was utilised for data analysis through the qualitative software Quirkos (https://www.quirkos.com/index.html). This software allowed for transcripts to be read and the text to be considered for the multiple meanings which were inherent in the body of the interview text [39]. Text segments containing meaningful themes relating to study objectives were assigned to categories by a member of the research team (CAKF). Throughout the analysis process, a hierarchy of categories was then established to show various relationships between categories [39]. Continued revision of the categories and emerging themes then took place with the Point searching for sub-topics, contracting points and new insights into each category. To demonstrate the emerging themes and categories identified during analysis, tables under each category have been included in the results section with illustrative quotes for each sub-topic. Demographic data (e.g., age, screening status etc.) was collected to describe the characteristics of the sample. Fifteen per cent of interview data were coded independently by two research team members investigators (CAKF, KAM).”

Point 5: Methods: Can the authors (circa lines 125) provide any data on agreement between two coders?

Response: The following has been added (though circa line 182) with an inter-rater reliability of 87% agreement reached’

Point 6: Methods: Can the authors describe any theoretical framework used to analyse the data? It is not clear how the themes were identified. Were the data from patients analysed separately to those from providers?

Response: As this was inductive, pragmatic research, no theoretical framework was used to analyse the data and we believe the description provided (lines 170-182 and copied above)  sufficiently details how the themes were identified. The methods section is already quite lengthy, therefore we have not added any additional information other than the following in line 187 to indicate that the provider and patient data were analysed separately: ‘which analyzed provider and female participant data separately

Point 7: Table 1. This table is unclear. Can the authors reorganise to show the patients and providers separately? Currently, the “stakeholders” are described under a table that is entitled “obese women”. The difference between the two is not clear. Do the authors have information on average/median length of clinical experience among the clinical stakeholders?

Response: We agree the table was unclear and have now corrected the title to read ‘Demographic characteristics of obese women and providers’ and have clearly delineated the difference between the two with use of better headings and shading.

Yes, we have information on the median length of service and rage, this has been added to Table one, thank you for this suggestion

Point 8: This Point found the results section very confusing.  I would like more differentiation between presentation and discussion of patient and provider results. It is not clear which of the themes were identified from patients and which from providers.

Response: This confusion is noted therefore we have done the following;

1) Reworded the introductory paragraph of the results section to read as follows: ‘Emergent themes from the obese female participant data were sorted into three main categories, all containing several sub-categories: 1) Obese women experience multiple issues around screening; 2) Body image concerns among obese women impact on screening attendance; 3) Negative experiences for obese women during screening can act as a barrier to future screening. Themes emerging from the provider data were sorted into two categories, again containing several sub categories: 1) Provider reported experiences with obese screeners; 2) Providers do not see obesity as being a barrier to breast screening. Examples of obese female participant and provider excerpts for each of these emergent themes can be found in Tables 2–6.’

2) Updated all theme names (and corresponding subheadings) to clearly differentiate between the two groups of results i.e. all previously titled ‘participant’ themes now updated to include ‘obese female’

Point 9: Results: The first “theme” (multiple encouragers and barriers…) seems more like a finding than a theme?  It seems many themes are discussed in section 3.1 and table 2. It is not clear how the themes presented in this section are different from the themes presented in the other sections? Thus, it was not clear to me how the items discussed in Tables 2,3, and 4 are different, and why they were separated as such?

Response: We agree this was a poorly named theme as it did appear to be an overall finding which led to lack of differentiation between the themes discussed in the ensuing sections. We have now updated the first theme to read Obese women’s understanding and awareness of screening can affect participation’ to more accurately reflect the themes findings and to delineate it from being an overall finding. This now means that the these presented in Tables, 2 3 and 4 are delineated.

Point 10: Line 150.  The “unhealthy behaviours” are not mentioned in the methods section. Are these investigator assessed or self-reported/identified?

Response: These unhealthy behaviours were self-identified by the female participant as demonstrated in excerpt 1.4, however to provide clarity the sentence has been updated to read as follows: ‘Only one participant was aware of her increased risk of post-menopausal breast cancer making reference to her weight (and her personal history of smoking and drinking), and her heightened need to screen (excerpt 1.4)’

Point 11: Is there a way for authors in the tables to distinguish which items refer to barriers and which facilitators?

Response: As this theme/table has now been updated (not referred to as barriers and facilitators, this comment is now not relevant.

Point 12: The authors identify several body image concerns discussed by obese participants in the interviews. Are these the “result” of their obesity, or do non-obese women have the same concerns? This may be worth discussing in the context of the literature in the discussion section.

Response: This is an interesting point as we know that body image concerns are common among women of all shapes however there is no literature  to the authors knowledge around body image concerns and mammography among normal weight women, though there is some among specific sub-populations like Aboriginal, women. We have therefore added the following to the discussion, paragraph 2 Further understanding of whether these body image concerns are primarily due to larger size or whether it impacts on mammographic screening by women of all body size is needed’.

Point 13: Section 3.4 The first two paragraphs seem to discuss redundant themes.  Can the authors merge and shorten?

Response: Thank you for this comment, we have now merged and shortened these two sub themes and have (and updated table 5 accordingly). ‘Impacts on the procedure as well as screening length due to the size of the women being screened. These were common reoccurring themes with the majority of providers during their interviews, which sometimes led to a focus on the task, rather than the patient.  For example, limitations with available equipment caused issues during mammogram with the ability of the machine to reach the desired compression reported to be compromised for patients with larger breasts (excerpt 4.1). Positioning of the patient was reported as being problematic, often due to the patient having a large stomach and the need for an anatomically inferior view (excerpt 4.2). Size also led to an increased number of images being required (excerpt 4.3) which in turn increased the ‘manhandling’ required, putting patients at increased risk of adverse events such as splitting the skin under the breast (excerpt 4.4). A typical outcome of these issues would be for patients having to subsequently return to complete their mammogram, sometimes at another clinic. The number of images required was not the only factor that could increase the mammogram length. Obese patients reportedly can become short of breath, limiting the amount of time they can stand, meaning that breaks also need to be taken to give the patient a rest (excerpt 4.5).’

Point 14: Line 283: suggest revising the word “distain” to “negativity,” or similar. The very negative language appears implicitly fatphobic?

Response: We agree this could be construed as being fatphobic and have updated ‘disdain’ to now read ‘negativity’

Point 15: The authors discuss the need to understand the communication needs of obese women. However, I think it’s also worth discussion that providers need to be trained to conduct screening with obese women, including sensitivity training.  

Response: This is an excellent point (and an already identified area for future research), therefore we have added the following to the discussion (paragraph 4): ‘Further, development of practical strategies on how to conduct screening with obese women, including sensitivity training could reduce the negative experiences of obese women as well as improve screening length times’

Point 16: I would like to see the authors mention next steps from this work in the conclusions section. What changes to clinical practice can be recommended as a result of this work? What gaps still exist?

Response: Thank you for this suggestion, We have now amended/added the conclusion so that it now reads as follows form circa line 600:changes to clinical practice should be considered including open identification of weight prior mammogram appointments, referral of obese women to services with equipment appropriate for larger women, larger women advised of likely longer screening time prior to their mammogram and education for mammographic staff. Further research should also be conducted to create a comprehensive picture of breast screening patterns among obese women, using routinely collected data that can be leveraged to inform appropriate community and screening service-based interventions. Impacts on attendance to healthcare services in Australia in general may also be needed given obese women’s increased risk of multiple chronic diseases, not just breast cancer’

Reviewer 2 Report

Minor comments:

Title: It is not clear enough about the content. I believe that including the words "qualitative study" and "obese women" would help to better understand the content of the manuscript.

Material and methods: participants were aged 45-80 years old. In Europe it is recommended to screen for breast cancer between 50-69 years. Although the age extension does not affect the conclusions, I advise to explain why the age range has been extended.

Results: The text can be reduced in some paragraphs without affecting the understanding of the results.

Table 1: introduce dispersion measures for age.

Table 2-4: specifying whether the comment was made by a woman participating or not participating in the screening will improve the understanding of the comment

Table 2: the following comment is not understood: “*Participants are from the obese women's group”. According to Table 1 all the participants are obese.

Conclusions: the sentence: "In turn this could be putting them at risk of increased breast cancer morbidity and mortality" does not correlate with the results of the study. I advise to remove it.

Author Response

Response to Reviewer 2:

We thank the Point for their time in looking at this paper and providing such useful feedback to help us improve the manuscript. Please see our responses to the comments below.

Point 1: Title: It is not clear enough about the content. I believe that including the words "qualitative study" and "obese women" would help to better understand the content of the manuscript.

Response: Thank you for this suggestion, the title has been updated and now reads as follows: Double Discourse: Qualitative Perspectives on Breast Screening Participation among Obese Women and Their Health Care Providers

Point 2: Material and methods: participants were aged 45-80 years old. In Europe it is recommended to screen for breast cancer between 50-69 years. Although the age extension does not affect the conclusions, I advise to explain why the age range has been extended.

Response: Thank you for this suggestion, the following sentences have been added: Women from age 45–80 years were targeted. While women are only invited for biennial mammograms in Australia from 50-74 years, women are able to self-nominate for a free mammogram from 40 years, as are women older than 74 therefore may have had recent mammogram.’

Point 3: Results: The text can be reduced in some paragraphs without affecting the understanding of the results.

Response: Thank you for this suggestion. Section 3.4. Provider reported experiences with obese screeners have been significantly reduced based on reviewer one's feedback. Further, words have been trimmed throughout the results section to reduce the length.

Point 4: Table 1: introduce dispersion measures for age.

Response: This has been done, with the range (47-75) now added

Point 5: Table 2: the following comment is not understood: “*Participants are from the obese women's group”. According to Table 1 all the participants are obese.

Response:

We agree the table was unclear and have now corrected the title to read ‘Demographic characteristics of obese women and providers’ and have clearly delineated the difference between the two with use of better headings and shading. Also, we have:

a) Reworded the introductory paragraph of the results section to read as follows: ‘Emergent themes from the obese female participant data were sorted into three main categories, all containing several sub-categories: 1) Obese women experience multiple issues around screening; 2) Body image concerns among obese women impact on screening attendance; 3) Negative experiences for obese women during screening can act as a barrier to future screening. Themes emerging from the provider data were sorted into two categories, again containing several sub categories: 1) Provider reported experiences with obese screeners; 2) Providers do not see obesity as being a barrier to breast screening. Examples of obese female participant and provider excerpts for each of these emergent themes can be found in Tables 2–6.’

b) Updated all theme names (and corresponding subheadings) to clearly differentiate between the two groups of results i.e. all previously titled ‘participant’ themes now updated to include ‘obese female’

Point 6: Conclusions: the sentence: "In turn this could be putting them at risk of increased breast cancer morbidity and mortality" does not correlate with the results of the study. I advise to remove it.

Response: Noted and removed
